# Memory Offloading for Remote Attestation of Multi-Service IoT Devices [note 1]

**DOI:** 10.3390/s22124340

**Published:** 2022-06-08

**Authors:** Edlira Dushku, Jeppe Hagelskjær Østergaard, Nicola Dragoni

**Affiliations:** DTU Compute, Technical University of Denmark (DTU), 2800 Kgs. Lyngby, Denmark; jeppe.jho@gmail.com (J.H.Ø.); ndra@dtu.dk (N.D.)

**Keywords:** IoT security, remote attestation, dynamic attestation, multi-service IoT device, memory offloading

## Abstract

Remote attestation (RA) is an effective malware detection mechanism that allows a trusted entity (Verifier) to detect a potentially compromised remote device (Prover). The recent research works are proposing advanced Control-Flow Attestation (CFA) protocols that are able to trace the Prover’s execution flow to detect runtime attacks. Nevertheless, several memory regions remain unattested, leaving the Prover vulnerable to data memory and mobile adversaries. Multi-service devices, whose integrity is also dependent on the integrity of any attached external peripheral devices, are particularly vulnerable to such attacks. This paper extends the state-of-the-art RA schemes by presenting ERAMO, a protocol that attests larger memory regions by adopting the memory offloading approach. We validate and evaluate ERAMO with a hardware proof-of-concept implementation using a TrustZone-capable LPC55S69 running two sensor nodes. We enhance the protocol by providing extensive memory analysis insights for multi-service devices, demonstrating that it is possible to analyze and attest the memory of the attached peripherals. Experiments confirm the feasibility and effectiveness of ERAMO in attesting dynamic memory regions.

## 1. Introduction

Nowadays, tiny electronic devices are increasingly deployed online and exist almost everywhere: from smart homes and smart cities to smart industrial systems. The recent Internet of Things (IoT) revolution is empowering the so-called *multi-service* devices that can provide multiple functionalities, for instance, a set of sensing capabilities. Such multi-service devices are supported by various well-known development kits (e.g., ST [1], Arduino [2]) that simplify the development of multi-sensor solutions. With the increasing number of services provided by IoT devices, smart infrastructures are expected to offer large-scale applications powered by an enormous number of interacting services (e.g., autonomous vehicles, traffic management services). However, with such exponential growth, IoT devices pose serious security and privacy concerns mainly due to their poor security design. IoT security is not prioritized throughout the product development process, resulting in devices with weak passwords, unencrypted communication, and or vulnerabilities to certain attacks. Actually, critical vulnerabilities can even be found on medical devices, such as insulin pumps not authenticating the legitimacy of the user or blood refrigeration units being protected by only a hardcoded password [3]. Further, traditional security solutions are not suitable for IoT devices because of their low-cost design and deployment in remote environments with limited physical accessibility.

Remote attestation (RA) has been proposed in the literature as a viable security solution for detecting malware presence remotely by providing a lightweight security mechanism through remote access. RA allows an external trusted party called the Verifier to verify the integrity of a potentially untrusted device called the Prover. Typically, existing RA protocols aim at detecting malware presence in program memory. This is done by computing the hash or the checksum of the program binary, which then an external Verifier can compare with the expected valid hash values. However, the hash computation is impractical for the attestation of dynamic memory regions (e.g., RAM) that change at runtime.

### 1.1. Motivation

IoT devices are exposed to a wide variety of rapidly evolving attacks, in particular, runtime attacks, where attackers exploit buffer overflow vulnerabilities or leverage Return-Oriented Programming (ROP) [4] to hijack the execution flow of a running program without injecting a new malicious code in the device. This has lead researchers to propose different dynamic RA approaches (e.g., [5,6,7]) to detect compromised devices. Such dynamic RA approaches use complex algorithm to trace the runtime execution flow of IoT devices, introducing high overhead or external hardware to the Prover. However, some attacks still remain undetected. For instance, Data-Oriented Programming (DOP) attacks [8] can compromise variables without deviating the control-flow execution of the running software. Additionally, existing RA schemes do not cover the entire memory of IoT devices, exposing the devices to mobile attacks [9,10] that can relocate themselves during attestation and exposing devices to be purposefully misconfigured, compromising their integrity without detection. This is especially crucial for multi-service devices. Indeed, the integrity of such devices depends also on the integrity of any attached external peripheral devices. For example, if an adversary successfully alters the configuration of a peripheral temperature sensor to provide an inaccurate representation of the temperature, any internal process that relies on this data, as well as any other system to which this inaccurate data is propagated, may behave in an unexpected way. As a result, in order to accurately verify the device, all attached peripheral devices must also be verified.

In order to verify the runtime integrity of IoT devices, including the verification of the external peripheral register, we rely on *memory offloading*. Specifically, rather than building complex RA algorithms for resource-constrained IoT devices, we propose offloading Prover’s memory to a powerful platform with more resources and computational capabilities, which will subsequently execute the attestation. The approach leverages the opportunities offered by the emerging Fog computing paradigm [11], where a layer of distributed powerful computing entities (i.e., Fog nodes) can enable the deployment of RA services. This allows the Verifier, deployed and running on a Fog node, to perform a much more accurate remote attestation of the IoT devices (Provers) the Verifier is responsible for.

### 1.2. Contribution

This paper brings the following main contributions to the research field of remote attestation:The paper proposes a novel RA protocol (ERAMO—Effective Remote Attestation through Memory Offloading) that takes into account both static and dynamic memory regions of an IoT device and checks the integrity of all memory-mapped peripherals.This paper successfully implements and evaluates secure memory offloading as a means for enhancing remote attestation. ERAMO has been implemented and evaluated on an ARM Cortex-M33 based microcontroller, leveraging the security features provided by ARM TrustZone.The paper expands the remote attestation procedure to attest multi-service devices by using offloading. We provide comprehensive memory analysis details, demonstrating that the flash memory, RAM, boot ROM and peripheral device registers adhere to some patterns.The paper evaluates the protocol on various metrics, e.g., transmission time, data authentication time, energy consumption. The conducted experiments confirm the feasibility of ERAMO and demonstrate that offloading technique increases the RA effectiveness in attesting dynamic memory regions.

This paper is an extension of our previous preliminary work entitled “ERAMO: Effective Remote Attestation through Memory Offloading” presented in 2021 at the IEEE International Conference on Cyber Security and Resilience (CSR) [12]. Apart from minor extensions and rephrasing in all the sections, we have extended the contribution of the paper mainly in two ways:The proof-of-concept implementation is significantly enhanced by considering a multi-service device with two sensor nodes. We present comprehensive details about the implementation setting, including the memory layout of the sensor nodes. In addition, we provide extensive analysis of the memory patterns of multi-service devices.The evaluation is extended to include additional results for energy consumption measurement which are important and were absent in the previous work.

In particular, we have included the following novel material in this paper:Section 3.3 has been added to include the most recent RA protocols for attesting IoT services. These state-of-the-art IoT service RA protocols are particularly relevant to the current paper.A detailed background related to ARM TrustZone functioning and working methodologies has been added in Section 4.Section 9 provides exhaustive implementation details for multi-sensor devices. In addition, we include the detailed memory analysis of the new approach.Section 10.1 provides with energy consumption evaluation. The evaluations prove the lightweight nature of the proposed scheme.Section 12 is extended to provide a comprehensive discussion about the memory locking, as a promising mechanism in improving the security and performance of memory offloading approach.New figures have been added throughout the manuscript to enhanse the representation of our proposed mechanisms.

### 1.3. Organization

The remainder of this paper is organized as follows. We explain the problem statement in Section 2. Section 3 presents different RA approaches and compares ERAMO with the existing RA schemes. Section 4 provides background knowledge on ARM TrustZone technology. The paper describes the system model in Section 5 and explains the adversary model in Section 6. Next, the paper presents security requirements in Section 7 and the protocol details in Section 8. The implementation details are presented in Section 9 the performance evaluation in Section 10. We highlight the protocol limitations in Section 11. Finally, we present a discussion in Section 12 and concluding remarks in Section 13.

## 2. Problem Statement

Consider an attacker that discovers and exploits a program vulnerability such as a buffer overflow. By leveraging Return-Oriented Programming (ROP) technique [4], the attacker alters at runtime the execution flow of legitimate code already loaded on the device’s memory to produce a malicious operation. Additionally, the attacker can use Data-Oriented Programming (DOP) technique [8] to compromise variables’ values and manipulate data pointers. Such attacks are common in IoT as resource-constrained IoT devices are exposed to many well-known vulnerabilities e.g., format string and integer overflow.

The dynamic RA protocols in the literature (e.g., [5,6,7]) which aim to detect control-flow attacks rely on tracing the software execution inside an IoT device and representing each execution flow as a single hash value. Since these approaches detect the control-flow subversion, they do not detect data attacks which do not maliciously deviate from the legitimate control-flow executions. The RA schemes presented in [9,10] aim to detect mobile adversaries, which during the attestation, relocate to different memory blocks of a memory region (i.e., memory blocks that comprise the program memory). However, the existing RA schemes do not attest all the memory regions of an IoT device. Thus, at the attestation time, a mobile adversary could also move to the unattested memory regions and relocate again on the original memory once the RA procedure has finished.

In the context of the attacks described above, we propose a new protocol that uses memory offloading to shift the attestation from low-end devices to nearby devices with more powerful computational capabilities. This approach is aligned with and leverages the emerging Fog computing paradigm, which extends the Cloud by bringing computational resources next to IoT devices [11].

## 3. Related Works

This section summarizes the single-device and dynamic state-of-the-art RA protocols in the IoT domain.

### 3.1. Remote Attestation Overview

RA approaches are generally classified into three main categories: software-based, hardware-based and hybrid approaches. Software-based schemes (e.g., SWATT [13], Pioneer [14]) do not make any hardware assumption and purely rely on the strict execution time of the RA protocol. Despite their advantages, software-based RA schemes do not provide strong security guarantees [15,16]. Hardware-based schemes (e.g., [17,18]) use a tamper-resistance hardware module as a Trusted Execution Environment (TEE). While hardware-based designs provide strong security guarantees, they are not suitable for low-cost resource-constrained IoT devices. To provide lightweight secure RA protocols, hybrid designs (e.g., SMART [19], TrustLite [20], TyTan [21]) rely on minimal hardware changes to ensure that the RA protocol and associated authentication keys cannot be tampered with. All these schemes perform attestation on a single device. Collective attestation schemes (e.g., SEDA [22], SANA [23], SHeLA [24], PADS [25,26], PERMANENT [27]) aim to provide scalable RA solutions that attest efficiently large-scale IoT networks.

SMARM [9] aims to detect mobile adversaries that, during attestation, relocate to different memory blocks of the program memory. SMARM uses a probabilistic approach to compute memory measurements in a random order, which cannot be predicted by malware. However, the probabilistic random (shuffled) measurements increase the attestation time. ERASMUS [10] is a non-interactive RA protocol that allows the Prover to self-initiate the attestation procedure at pre-defined times. The attestation results are stored locally, and the Verifier retrieves a set of attestation results. The sequence of the attestation results allows the Verifier to detect mobile adversaries that may leave or get relocated during attestation.

### 3.2. Dynamic Remote Attestation

While the aforementioned RA schemes perform only static attestation, dynamic RA schemes aim to attest dynamic data memory. C-FLAT [5] is the first dynamic RA protocol for resource-constrained devices, and it focuses on detecting control-flow attacks. C-FLAT relies on software instrumentation to trace the execution of a running software and generates an accumulative single hash value for each execution flow. At the verification phase, the Verifier compares the generated hash value with a set of expected legitimate values to determine whether the device is trustworthy or not. C-FLAT is implemented in a TEE such as TrustZone. However, C-FLAT introduces a high overhead because at runtime each instrumented code instruction is intercepted and redirected to the TrustZone secure world. LO-FAT [6] enhances C-FLAT by replacing software instrumentation with a hardware module, implemented on an external FPGA, which intercepts the executed instructions at runtime. Likewise, ATRIUM [7] extends C-FLAT and LO-FAT by attesting both executed instructions and the control-flow. However, these schemes detect control-flow deviations and do not consider data attacks which leverage DOP technique [8] to corrupt data variables without altering control-flow information. LiteHAX [28] aims to detect both control-flow and data-attacks. However, LiteHAX detects only the memory operations *load* and *store*, thus, it works only on RISC-based architectures.

### 3.3. Service Attestation in IoT Swarms

Some recent RA protocols in the literature consider the attestation of IoT devices that contain one or more services (also called as modules) [29,30,31,32].

DIAT [30] aims to perform the attestation of modules in the embedded devices of an autonomous collaborative system. For each pair of interacting IoT modules, DIAT performs control-flow attestation and authenticates the exchanged data between each pair. In this way, DIAT ensures that the data sent from one module to another has not been maliciously changed. RADIS [31] attests a group of interacting services that compose a distributed IoT service. In order to detect malicious services that impact the behaviour of other legitimate services in the network, RADIS performs the control-flow attestation of the entire distributed service. SARA [33] aims to attest distributed IoT service communicating by a publish/subscribe scheme. By using logical vector clocks, SARA allows the verifier to construct a historical graph of the occurrence of service interactions and identify the maliciously influenced provers. While SARA relies on static attestation, ARCADIS [34] extends SARA by performing control-flow attestation. In particular, ARCADIS detects IoT devices that have (directly or indirectly) been maliciously influenced by runtime attacks on asynchronous distributed IoT services.

### 3.4. Memory Offloading

Beside RA protocols, some works within the field of offloading are of interest. In particular, CloneCloud [35] allows a resource-constrained mobile device to offload its execution threads to a clone of itself operating in a virtual machine with more computational capabilities.

In the context of remote attestation it will, however, be more useful to gather an accurate clone of the device memory, on which the memory forensics can be performed, rather than replicating the functionality of the device. Additionally, certain security guarantees not considered by offloading techniques must be provided by RA designs, as they are intended to be used on potentially malware-infected platforms. Due to these differences in security requirements and their purpose, the works within the field of offloading are not directly applicable.

The possibility of offloading in the RA context is first mentioned in RAaS [36], where RA is proposed as a cloud service. While this proposal is mostly focused on increasing the efficiency of the protocol and reducing its associated downtime, the proposal has not been implemented and evaluated. Furthermore, we want to investigate whether offloading the memory is feasible and how this technique can be used to improve the effectiveness of remote attestation.

### 3.5. Discussion

Table 1 summarizes the works discussed so far. In short, the static RA approach does not consider runtime attacks, while recent dynamic approaches are limited to detecting control-flow attacks. Thus, even if the approaches are combined, data attacks will still remain undetectable.

This paper proposes a new protocol (ERAMO) aimed at addressing the limitations of current attestation designs. Instead of relying on the control-flow attestation or checksum/hash comparisons, we propose to transmit the entire memory to the Verifier. This allows the Verifier to employ sophisticated methods of attesting the dynamic memory (e.g., the open-source Volatility memory forensics framework [37]), while expanding the protocol to also cover the Prover’s peripherals (e.g., ADC and I2C configurations). ERAMO design is based on ARM TrustZone [38], which is a hardware-enforced isolation method. However, as opposed to other hardware-based methods, TrustZone is built into the CPU, providing a TEE without the need for external specialised hardware.

## 4. Background

This section summarizes some relevant background information regarding the concepts and technologies used in this paper.

### 4.1. Runtime Attacks

Runtime attacks exploit a memory corruption vulnerability to manipulate the control-flow or data-flow of a program. The most common types of runtime attacks are (1) code-injection attacks and (2) code-reuse attacks. Figure 1 illustrates a code-injection attack on a Control-Flow Graph (CFG) of a benign program, where CFG represents the compiled program’s code including all possible legitimate control-flow executions of the program’s statements (Nodes 1-5 in Figure 1). In particular, in code-injection attacks, the adversary exploits a vulnerability (Step 1 in Figure 1), directly injects malicious code (node X) into the memory space of an application (Step 2), and manipulates the control-flow to point towards the malicious code (Step 3). Code-injection attacks can be prevented by enabling security policies such as Data Execution Prevention (DEP) which guarantees that writable memory locations are non-executable. However, DEP does not prevent the attacker from reusing the legitimate code already loaded on the device.

In code-reuse attacks, the adversary exploits a memory corruption vulnerability and compromises the intended behaviour of the program by manipulating the execution order of the legitimate sequences of the code (called gadgets) already present on the device. For instance, in Figure 2, the code-reuse attack diverts the control-flow from (3,4) to (3,2) to execute code in the other branch. The most common variants of code-reuse attacks are Return-Oriented Programming (ROP) [4] and Jump-Oriented Programming (JOP) [39,40].

### 4.2. Fog Computing

Fog is a relatively new architecture that shifts computation from the cloud to the network edge, closer to the data producers [41,42]. The computational, networking, storage, and acceleration elements in a fog computing paradigm are known as fog nodes. Figure 3 illustrates a logical hierarchy of the computing resources in an IoT system, consisting of three main layers:Sensors and actuators: Sensors and actuator devices are the physical things that produce data. These IoT devices are heterogeneous with different processing capabilities, ranging from very simple devices with minimal resources to more powerful devices that can support wired or wireless protocols, such as BTLE, ZigBee, USB, Ethernet, etc. Many IoT devices can be associated with a single fog node.Fog nodes: Fog nodes at the edge are often used for sensor data collection, data standardization, and command/control of sensors and actuators. Fog nodes in the higher layer typically focus on data filtering, compression, aggregation, and turning the data into knowledge.Cloud: The traditional backend cloud remains an important part of a fog computing paradigm, performing tasks that are not completed by the fog devices.

Fog computing, that distributes computational resources to the network edge, is a promising paradigm for dealing with the performance and network congestion issues brought by the exponentially growing volume of data created by connected devices. Further, fog computing addresses performance, security, bandwidth, reliability issues in the systems where cloud-only solutions are unsuitable [43].

Many major cloud providers have adopted the fog computing paradigm, providing edge-oriented frameworks such as Azure IoT Edge [44], AWS Greengrass [45], EdgeX [46], etc. These frameworks can be designed to filter device data and only send necessary information to the cloud.

### 4.3. Arm Trustzone

ARM TrustZone technology is centered around splitting the system into a secure and a non-secure (also called normal) domain and enforcing this separation by hardware. Both domains are completely hardware separated and have their own privileges. At any given time, the processor will operate either exclusively in the secure or non-secure domain.

While TrustZone for ARMv8-A (Cortex-A) and ARMv8-M (Cortex-M) are very similar, both being built on the same foundation, there are slight differences. TrustZone for ARMv8-M extends the architecture with new features and further separates the secure parameters from the rest of the application. There are also changes to the state transition between secure and non-secure, as the ARMv8-M performs the transition in hardware, unlike the ARMv8-A, which employs secure monitor software [47]. With the separation done in hardware, new opportunities arise to constrain the OS to the normal domain, while the secure methods can reside in the secure world without the need for OS interference such that we do not rely on a the OS being secure.

A visualisation of the differences between ARM Cortex-A and ARM Cortex-M is depicted in Figure 4. ARM Cortex-A performs the transition to the secure world through the secure monitor, which acts as a context switch triggered by a hardware interrupt or the software instruction SMC. This transition will set the processor’s Non-Secure (NS) bit, readable from the Secure Configuration Register (SCR) and propagated throughout the system. In short, the security state in Cortex-A is determined by this bit.

On the other hand, the Cortex-M does not have the secure monitor, it does not need to go through any transitional mode and does not have an NS bit deciding its state. Instead, Cortex-M’s security state is entirely decided by whether the code being executed is in a secure region or a non-secure one. As shown in Figure 4b, the transition is possible anywhere in the application. The software can be brought to the secure world by an interrupt request (IRQ) or with a function call and return in the same manner whether the software is operating in thread or handler mode. Avoiding the transitional state also improves the transition speed in Cortex-M over the Cortex-A [48].

When discussing TrustZone henceforth it will implicitly be the Armv8-M (Cortex-M) version unless stated otherwise, as this is the platform available for the low-end devices relevant in the context of remote attestation.

The processor will either run in a secure or non-secure state with access to either secure or non-secure firmware, data, peripherals, memory and resources. This is visualised in Figure 5 that shows how each area is either designated to the trusted or the non-trusted domain. The memory space is separated into secure and non-secure addresses. The non-secure addresses can be accessed by all the software on the system, while the secure memory space is split into secure and non-secure callable (NSC). Secure addresses are only accessible by secure software, while NSC are addresses reserved for instructions allowing the software to transition from the non-secure state to secure [38]. Non-secure code calling the a secure functions directly will result in an exception, so to avoid this non-secure software can call permitted secure functions through the NSC section. This NSC section will be addressable by the non-secure software through using its veneer table, which will maps the entry-points to the secure domain.

It should also be noted that TrustZone itself does not ensure that the system will be secure. TrustZone provide the means to which a secure system can be developed, but it has to be implemented correctly with respect to cybersecurity principles. If the software is not properly written and the separation between secure and non-secure states is not properly implemented, an attacker may still able to exploit this and attack the system. As code gets more difficult to evaluate as it grows, it is recommended to keep the secure code as minimal as possible and keeping the amount of entry-points few, while performing most operations in the non-secure state. Several critical vulnerabilities have been discovered in popular TrustZone implementations spanning over several systems and OSs [49]. This vulnerabilities range from compromising the kernel using buffer overflow exploits to lack of isolation between secure and non-secure due to debugging channels. It should also be noted that while TrustZone may be secure in software, a physical adversary could possibly still attack the system. Fault injection attacks have been successfully performed on several ARMv8-M TrustZone microcontrollers [50].

## 5. System Model

We consider an IoT system which adopts Fog computing paradigm [11]. In this system, an untrusted resource-constrained IoT device interacts with a nearby powerful device named Fog node. To design a RA protocol in this setting, we consider the presence of the following two entities as shown in Figure 6:The Prover (Prv) is an untrusted IoT device. We assume the Prover to be a multi-service device, e.g., a multi-sensor IoT device that provides a set of sensing capabilities using external peripheral. This device can be infected by malware or can be misconfigured as a result of previous attacks. Prover’s memory consists of a set of *memory regions* as shown in Figure 7. Each memory region can be seen as a set of smaller units called *memory blocks*.The Verifier (Vrf) is responsible for checking the integrity of the Prover. In our system model, the Verifier is a fog node resided next to the IoT device. For simplicity, we consider a system with only one layer with fog devices, assuming that a fog node is a powerful device that has all the required resources and computational capabilities to perform complex operations. Alternatively, the Verifier’s task can be distributed in a set of hierarchical fog nodes where some degree of analysis is done by the fog node next to the IoT devices, while the other part of the task is performed by the nodes at the higher layers (as depicted in Figure 3). Besides, the Verifier has the required resources to adopt advanced security and trust techniques (e.g., it is equipped with a Trusted Platform Module (TPM) [51]); thus, it is assumed to be trusted. Additionally, aligned with other RA schemes in the literature, we assume that the Verifier knows in advance the legitimate program binaries and has the ability to detect irregularities in dynamic memory or peripherals of the Prover. The Verifier randomly initiates the attestation on the Prover, after which it can perform memory forensics techniques to determine the Prover’s integrity.

The Verifier initiates the attestation by sending a request to the Prover (Step 1 in Figure 7). After obtaining the attestation request, the Prover copies the content of its entire internal memory associated with the device application (Step 2) and offloads it to the Verifier (Step 3). Upon receiving the Prover’s memory, the Verifier will perform the verification (Step 4) to check the Prover’s trustworthiness. The verification process includes two main parts: a comparison of the static memory (e.g., flash memory) with the legitimate program binaries known in advance and a detailed investigation of the transferred data memories (e.g., SRAM).

## 6. Threat Model

In the following, we define the adversarial capabilities w.r.t. the system model described in Section 5.

### 6.1. Adversarial Actions

In line with the adversary model described in [36,52,53], we consider an adversary with the following capabilities.

Software attack: A software adversary compromises the Prover’s program memory by injecting and executing malicious code. Additionally, this adversary can exploit a software vulnerability to compromise data memory, for instance, by modifying variable’s value, corrupting control-flow pointers, data pointers. This can also be exploited to misconfigure internal or external peripheral to cause unintended device behaviour.Communication attack: The communication adversary can fully control communications between the Prover and the Verifier by forging, dropping, delaying, eavesdropping the exchanged messages.Mobile attack: A mobile adversary is a smart adversary that tries to avoid detection by deleting itself during the attestation time or relocating itself to different memory blocks or memory regions which have already been transmitted to the Verifier.Replay attack: An adversary precomputes a valid attestation response and sends this old legitimate response to hide an ongoing attack.

### 6.2. Attack Capabilities and Limitations

We assume that a remote software attacker can read and write arbitrary memory. In particular, the attacker is able to perform a runtime attack by exploiting a memory-corruption vulnerability. However, we assume that the adversary does not compromise the hardware-protected memory. Further, we assume that the adversary has computation capabilities to perform computations at runtime. The main goal of the adversary is to compute a valid attestation response to hide an ongoing attack and remain undetected. Thus, following the assumptions of other RA schemes [5,19,22], we rule out physical adversaries, Denial of Service (DoS), and Time-Of-Check Time-Of-Use (TOCTOU) attacks. While we do not consider TOCTOU attacks, we limit these attacks by transmitting complete device memory to a powerful Verifier that performs advanced analysis or historical comparison over the memory contents.

### 6.3. Defense Capabilities

In line with common assumptions of the state-of-the-art RA schemes, we assume the presence of two trusted components inside a Prover device.

Read-Only Memory (ROM). A ROM memory region contains the code of ERAMO protocol. The protocol code resided in this memory region cannot be tampered with by a software adversary.Secure key storage. A secure memory region stores the Prover’s keys. Only ERAMO protocol has read permissions in this memory region.

## 7. Security Requirements

Based on the adversarial actions described in Section 6, in the following we define the required security properties.

Integrity. The protocol should provide reliable evidence guaranteeing that the transmitted memory contents correspond to the Prover’s memory at the time of the attestation request.Authenticity. The protocol should provide verifiable evidence for the origin of the memory contents transmitted.Integrity of communication data. The protocol should ensure that any memory contents transmitted cannot be altered without it being detectable.Freshness. The protocol should ensure that any given response to an attestation request can be reliably linked to that request.

## 8. Eramo: Protocol Proposal

ERAMO protocol consists of three main phases: (1) Setup phase, (2) Attestation phase, and (3) Verification phase. In the following, we describe each phase in detail.

### 8.1. Setup Phase

A network operator guarantees the secure bootstrap of the software deployed on each Prover. Considering the limited capabilities of Provers, the Verifier and the Prover establish a shared symmetric attestation Message Authentication Code (MAC) key *k*. To prevent untrusted parties from using Prover’s key, the shared attestation key *k* is stored in a hardware-protected memory. Alternatively, a Prover can establish a secure communication channel with the Verifier by possessing an asymmetric key-pair (pk,sk) and knowing the Verifier’s public key. Note that the key management details are out of scope of this paper. The protocol description is independent of the key management, thus, the symmetric key usage can be easily replaced by an asymmetric key-pair. For simplicity, preserving our work’s generality, we assume that the Prover and the Verifier share a symmetric key *k*.

### 8.2. Attestation Phase

Figure 8 illustrates the protocol. To initiate the attestation, the Verifier generates a nonce *N* and sends it to the Prover (Step 1 in Figure 8). The Prover then relinquishes control to the RA protocol residing in the hardware-protected component. The Prover’s RA protocol reads the device memory contents *m* (Step 2) and computes a hash h=hash(m). Next, the Prover concatenates the computed hash *h* with the received nonce *N* and authenticates it by computing a keyed Hash Message Authentication Code (HMAC) over the obtained result s=HMAC(k,(h||N)). Finally, the memory *m* and HMAC *s* are transmitted to the Verifier (Step 3), which checks whether it corresponds to the transmitted data. The transmission may be split into smaller chunks, e.g., by authenticating individually memory blocks or regions. In that case, integrity, authenticity and temporal freshness must be ensured for each transmitted memory chunk, e.g., by adding a unique extra byte for each chunk or securely generating a pseudo-random number inside the Prover.

### 8.3. Verification Phase

The verification phase starts when the Verifier receives an attestation response from the Prover. By using the shared attestation key *k*, the Verifier checks the authenticity and integrity of the attestation result (Step 4). Assuming that the Verifier knows all valid combinations of memory *M*, the Verifier has the ability to determine whether a given memory *m* is in the set *M*. A powerful Verifier that is able to perform advanced memory forensics analysis (e.g., by using the open-source Volatility Framework [37]) can use the offloaded dynamic memory contents to provide a detailed attestation and precisely determine the Prover’s integrity.

### 8.4. Attested Device Memory

Figure 9 shows the attested memory regions verified by ERAMO protocol for a device with a flash memory and a memory-mapped peripheral region. A certain portion of the flash region allocates data memory, whereas the memory-mapped peripheral region contains both readable and write-only registers. All readable memory can be attested apart from the secure memory allocated to the trusted component performing attestation.

The inclusion of the aforementioned memory regions in the attestation result is crucial to ensure Prover’s integrity. In particular, the attestation of the memory-mapped on-chip peripheral address space guarantees that any on-chip peripheral in use works as intended, and an adversary has not altered the device’s peripheral configurations. These configurations may range from the ADC channel chosen, the I2C communication speed, or the internal timer setup. However, due to its dynamic status and configuration registers, this region cannot be attested by the comparison of hashes as each combination of configurations and status bits will produce a unique hash, resulting in a large amount of legitimate hashes. Additionally, registers may have unused or reserved bits with undefined read-values, which further complicates the hash verification. Therefore, this memory region should instead offloaded to the Verifier.

Furthermore, if a region of the flash/EEPROM is used for data, such as calibration values or network information, this region may also be verified through offloading. This data may change during runtime and may depend on the electrical characteristics of the specific device, and thus may not be verifiable through hashing. Assuming that the Verifier has some notion of what differentiates legitimate values of this region, the integrity verification of this region is possible through offloading.

#### Attestation of Multi-Service Devices

When the Prover is a multi-service device, its integrity also depends on the integrity of any attached external peripheral devices. The peripheral devices are typically not programmable but rely on limited interfaces, such as SPI, I2C, and UART, to read or write to their register contents. These registers, in the same manner as the on-chip peripherals, determine the peripheral configurations and contain their data. Consequently, the contents of these external peripheral registers should be verified to guarantee Prover’s integrity.

Before authenticating and offloading the registers’ content, first, the trusted component should read the registers. To accomplish this, an extra step is added to the attestation procedure within the trusted component. The Prover uses the peripheral interface (such as SPI or I2C) to read every accessible register on the external peripherals. The contents of these registers now reside within the trusted component and can be offloaded to the Verifier. The Verifier then verifies these external peripherals as it verifies the internal ones. The configuration bits and other data of the peripheral device can be evaluated by considering legal combinations or through more rigorous analysis.

## 9. Proof-of-Concept Implementation

We implement ERAMO protocol on a TrustZone-capable LPC55S69 running an IoT sensor application on FreeRTOS (https://www.freertos.org, accessed on 29 April 2022). To investigate an multi-service IoT system with different peripherals, we choose two sensors: a temperature sensor and a water sensor.

As an illustrative use case, we can imagine a temperature and water sensor IoT device in a server room. Every 100ms, the device reports the temperature and water level of the room. This IoT system may function as an anti-fire system, turning on the sprinklers if an increasingly high temperature is measured over a number of samples. Another device may be reading the water level the device reports, ready to turn off the servers and enable a countermeasure if a water leak happens. In such scenario, it could have disastrous if the temperature sensor node was hacked, reporting a high temperature to the sprinkler system, flooding the room, while in turn the water sensor was not reporting the water level.

### 9.1. Sensors

In our experimental setup, the temperature sensor is a digital sensor, namely Bosch BME280, to integrate a component with it’s own memory in the form of registers and to integrate some digital intra-device communication (as in I2C, SPI, I2S etc.). The water sensor was chosen as an analog sensor, a simple non-brand sensor that gives a voltage output depending on the water level. The peripheral sensors can be seen on Figure 10.

#### 9.1.1. Water Level Sensor

The water level sensor is an analog sensor, and it gives a voltage output depending on how immersed it is in water. It works as a resistor whose resistance varies by the water level. As it is purely analog, it has no internal memory and consequently cannot directly be attacked by malware. It is supplied with voltage and ground using two of the sensors pins, while the last pin will output a voltage depending on the water level. The output is equal to the supply voltage when the sensor is immersed in water and fully conducting, while it is equal to ground when the sensor is dry and is not conductive. This output voltage can then be converted to a digital value representing the water level using the microcontroller’s analog-to-digital converter (ADC). This PoC assumes a linear correlation between the analog voltage output and the water level.

However, the converted digital output still depends on the ADC configuration, which can be written to and read using the memory addresses of the ADC peripheral. The ADC must be configured to use the correct input, number format and conversion mode. In this case, it is configured to be triggered by a specific interrupt, which ensures that sensor readings are done at a regular interval. If these settings are misconfigured, the sensor may report incorrect readings, which would cause this device to behave incorrectly. Furthermore, the misreadings could propagate errors to other devices in the IoT network, whose behaviour depends on the accurate sensor readings. As a consequence, the memory associated with the peripheral will need to be offloaded and validated in order to fully attest the state of the device.

#### 9.1.2. Temperature Sensor—Bosch BME280

The Bosch BME280 is a digital humidity, pressure and temperature sensor. In the context of this PoC implementation, the sensor represents of a typical digital sensor in a IoT device.

While the sensor functions by either SPI or I2C communication, we choose I2C for this PoC implementation, considering that the transmission speed is not of importance. The datasheet strongly recommends that data is read using a burst read, which is specified as sending the slave address, the desired register address, after which the sensor starts reading out data from that address and on-wards until it no longer receives an ACK [54]. An example of this can be seen in Figure 11, where register 0xF6 and 0xF7 is read, after which the master stops the read by no longer replying ACK. The write is done in the simple manner of sending the slave address, the register address then the data. Note that correct configuration of the I2C peripheral must be verified in the same manner as for the water sensor. However, in this case the external memory residing on the digital sensor must also be verified to ensure correct readings.

The sensor consists of a variety of registers, 8 read-only registers containing raw humidity, temperature and pressure data. 3 read-write registers for configuration, 2 read-only for calibration data and 1 read-only each for the status, chip id and reset registers. The memory map of the sensor can be seen in Figure 12. Worth noting, is that the shown registers are not necessarily contiguous, there are registers in between the mapped register, which are not documented in the datasheet. By experimentation it is found that these registers can be read without issues and do not disturb the burst read of the memory.

Worth noting, is that the device starts in sleep mode and requires that the *mode*-register is set to normal mode and the oversampling registers are set to a nonzero value before the sensor will give accurate results.

If this sensor has a wrong value in its mode or oversampling register it would report back an incorrect result. As these registers are not checked by traditional remote attestation, a device could report back as being in a correct state with a legal memory hash, while having a corrupt sensor reporting incorrect temperature values to other IoT devices on its network. As external digital sensors are not directly linked to the Prover’s memory addresses, they will not be taken into account in a traditional remote attestation, even though their state is essential to the performance of the sensor node. If a mobile adversary exploits a vulnerability to change the state of this peripheral sensor, then deletes all traces of itself from memory, the invalid state cannot be detected by current methods of remote attestation.

### 9.2. Eramo Protocol

To isolate the RA protocol from the non-trusted device application, we use the Armv8-M release of ARM TrustZone. TrustZone is available on various platforms, and unlike other approaches to hardware-based attestation, does not require any external hardware. In general, TrustZone separates the device application into a trusted component allocated to the TrustZone secure world and a non-trusted component allocated in the TrustZone non-secure world. The LPC55S69 board setup with the sensors connected can be seen in Figure 13.

In ERAMO’s proof-of-concept implementation, the application is separated as follows:The trusted component includes the RA procedure and the LPC55S69 hash engine. Additionally, a section of the RAM and flash is allocated to the secure world. The attestation code and key are located in the secure flash, and the key is handled exclusively in secure RAM.The non-trusted component includes FreeRTOS, the IoT sensor application tasks, and associated interrupts and peripherals. The remaining RAM, flash, and any unused peripheral are also allocated to the non-secure world.

The trusted attestation protocol is initiated by calling it from the non-secure communications thread with the nonce *N* as an argument (fixed to a length of 8 bytes to prevent inputs of arbitrary length). The protocol performs the authentication using the LPC55S59 on-chip hash-engine because this significantly speeds up the hash computation. The chip supports SHA-1 and SHA-2 with a 256 digest (SHA-256). As SHA-1 has certain vulnerabilities [55,56], we use SHA-256 for hashing and the HMAC. To prevent key leakage, the hash engine is assigned to the secure world.

To illustrate the attestation of multi-service devices, the accessible registers of the BME280 external peripheral sensor are offloaded to the Verifier. The trusted attestation procedure on the device performs an I2C burst read on the BME280, resulting in the register contents being transferred to the trusted component’s I2C buffer. The burst read is performed using polling to not rely on the interrupts associated with the non-trusted component. The procedure then transmits the memory to Vrf in the same manner as its internal memory. While offloading the memory allows Vrf to verify with a variety of methods, the current implementation will confirm that certain bits of register 0xF2 (ctrl_hum), 0xF4 (ctrl_meas), and 0xF5 (config) (depicted in Figure 12) are configured as intended. The dynamic and reserved register are ignored. Additionally, the device-specific calibration values (0xE1-0xF0 and 0x88-0xA1) are read and logged such that they can be compared with internal values used in the device.

### 9.3. Memory Analysis of Sensor Nodes

We perform a memory analysis to investigate how the memory of the sensor node application is being utilised. This analysis will reveal whether the flash, RAM, boot ROM and peripheral device registers adheres to some structure, which can be predicted and used in the verification process. The device consists of boot ROM, flash memory, the temperature sensor registers and 4 different RAM regions SRAM0, SRAM1, SRAM2 and SRAMX.

The program flash is expected to remain static throughout the use of the application. The application is programmed into the flash and no part of the application is programmed to write to the flash. Consequently, the program flash can still be verified using hash values, since the Verifier would only have to know a limited amount of legal hashes corresponding to all legal program versions. On the other hand, the RAM of the application will have too many legal states to realistically keep hashes of. This does however not mean that there will not be patterns in the RAM and possible structural features which can be used to tell a legal use of RAM from an illegal one (one hijacked by malware). This analysis will therefore mainly involve the RAM used by the applications.

Before the analysis, the application was modified with tools to aid the analysis. FreeRTOS has internal tools for this cause, notably a timer for runtime stats using a hardware timer on the processor and the trace facility which adds additional structure members and functions to the FreeRTOS structures, enabling the extraction of information regarding the task/queue state, stack etc.

First, we used the FreeRTOS memory analyser, built into MCUXpresso IDE, to extract the memory addresses allocated to each task and queue. The addresses of these were noted for the following analysis. Using these it was possible to get information about where in the memory the different structures lie.

The analysis showed that the different FreeRTOS structures (the tasks and queues) consistently allocate the same memory spaces. This is due to the structures being allocated statically and before the scheduler has been started. As an example, for this version of the application, the UART Transmit Queue will always occupy address 0x3EC and the LED task address 0xD94 in RAM.

Then, using this information, the queues, tasks and the timer was plotted into the memory map in the address and information columns, Figure 14. As the application for the moment does not require much space in RAM, it is entirely located in SRAM0 (64 KB), with the heap start being set to 0x2000 0000 and the stack start (not to be confused with the FreeRTOS stacks, which are task-specific) being set to 0x2000 F000.

A problem may arise with dynamic allocation of FreeRTOS structures after the scheduler has begun. If a task is allocated dynamically, the Vrf will not know where in memory it is located, making the process of attesting the RAM and memory-locking even more complex. Some trusted scheme of exchanging the structure location could perhaps be derived, but this PoC implementation will instead be limited to statically created structures, such that Vrf has knowledge of the layout beforehand.

To perform the analysis, the memory had to be offloaded from the device. Complete readings of the memories were taken once an hour over the course of 16 h in order to get an accurate representation of the memory regions used. To further analyse the memory activity throughout these memory readings, we wrote code to calculate how many times each memory region changed over the course of the readings. This program compares each reading of the memory with each other: the first reading is compared with the second, the second with the third etc. Each block of memory (of 0x10 bytes) which changes value between two readings is noted. This process is done for each memory-region of the device, the RAM, flash, boot ROM etc. The results showed that the only memory regions changing during runtime were SRAM0 and the temperature sensor registers, meaning that the flash, boot ROM and SRAM1, SRAM2, SRAMX were static.

The SRAM0 information can be seen in the column “Changes” in Figure 14, where the memory is split into chunks of 0x100 bytes. The results show how the first 6KB (0x0000–0x1700) are being actively used by the device. There a some inactive blocks in-between, but most of it changes between each offload. After 0x20001700 there is a large gap without any activity until the last block of SRAM0, where the stack is located. Even though each FreeRTOS task has their own stack, the normal stack is still used by *main* and by any interrupts. The first memory space is occupied by the heap and the last space after the software timer hosts the scheduler. The scheduler is 236 bytes, each queue is 76 bytes plus storage area and each task is 64 bytes plus the size of its stack. Each task stack is set to 256 bytes, which is the default of this implementation’s FreeRTOS port.

In addition to researching the memory consumption of the whole system, it is also of interest to research the memory consumption of a scenario with a single task running. This is helpful to see if memory locking, a concept that will be elaborated in Section 12, can be applied to some regions of the system, while running tasks in others.

The experiment will be to observe how much of the memory space a single task would use, in a situation where all the tasks and RTOS structures have been created, but have been suspended. Meaning that they are allocated, but the scheduler will be ignoring them and their stack will remain static. To test this, the application was edited to suspend all tasks except the button task, the task which monitors the state of the MCU board’s button. The button-task was chosen as it does not use any queues, timers or rely on any other tasks. The offloading procedure and analysis was repeated for this new application and the results can be seen in the column *Changes (1 Task)* in Figure 14. It’s seen how the stack is still active and along with the top of the heap to addresses 0x640, where the UART Transmit task starts. While the queues themselves are not active, the memory surrounding them are. Additionally the memory allocated for the button task, the idle task, software timer and scheduler remains active.

The following registers on the external BME280 sensor 0xBD, 0xBE, 0xC1 and 0xF3 also changed throughout the offloads. The last register, 0xF3, is the *status* register of the sensor containing a bit indicating whether the sensor is busy measuring and a bit indicating whether the sensor is busy copying data. As this register is read-only and cannot be changed maliciously, it will be have to be ignored by the attestation process. The first three registers however are not described in the sensor datasheet as they are in the region between *calibration* (0x88 to 0xA1) and *id* (0x0D). Presumably these registers are used internally by the sensor, but it can’t be said what they are for and how they affect the temperature readings. If these registers are read-only they can be safely ignored in the attestation, however if they can be written to they may affect the device and could be an attack vector for transient malware creating lasting damage which is not detectable. To ensure safety it may be best to ensure that any undocumented registers of the sensor used in a device are read-only or do not have any negative effect if written to.

As a consequence to the observe patterns in the device’s memory, there are some opportunities and challenges for the attestation procedure. First of all, program flash, the boot ROM and the unused RAM are all static throughout the use of the program. This opens up the possibilities of memory locking while attesting these regions as they are not used by the application. The sensor values are also somewhat predictable with most of registers being verifiable depending on how they were initially set, however with certain areas which cannot be predicted as they lack documentation. The RAM does however pose some challenges to interruptible procedures. There are certain areas of the memory being used by FreeRTOS even when the system is idle, such as the idle task and any created software timers. The application also constantly uses the heap and the stack to handle system interrupts. It should however be noted that while these parts of the RAM are problematic for the procedure, only 10% of SRAM0 (about 6KB) is actually used by the application. The remaining SRAM0 and the SRAM1, SRAM2 and SRAMX is completely static and is therefore very predictable.

In conclusion, only the RAM in use by the application will give issues during offloading. This part of the RAM does however adhere to a predictable structure regarding the placement of FreeRTOS objects (tasks, queue, timers). While the content of these objects may be difficult to verify, the structure itself is predictable, as long as the objects are statically allocated. The program flash and boot ROM remain static and therefore verifiable by hash and the analysis of the BME280 sensor reveals that most of the registers remain static, thereby opening some possibilities of peripheral device verification.

#### Attested Memory and Device Integrity

The developed attestation procedure successfully offloads the RAM, the boot ROM, and the flash memory. Certain sections of the memory-mapped peripheral are write-only, thus, only the readable addresses are transmitted. For this implementation, the I2C peripheral is offloaded, and the Verifier verifies that the I2C interface is configured as intended by verifying the configuration (CFG) register, the interrupt settings (INTENSET), and the settings for the clock and timings (CLKDIV).

The Verifier successfully then verifies the boot ROM and the region of the flash memory containing the program binary by comparing the hashes, before moving on to the dynamic areas of the memory. Currently, no memory forensics is performed on the RAM, however we discuss its feasibility in Section 12.

## 10. Evaluation

The efficiency of ERAMO highly depends on the choice of hardware. The memory transmissions depend on the choice of communication and its transmissions speed. The time required for authentication depends on Prover’s computational capabilities and its available hardware to assist with the process.

We conducted the experiments, and the runtime measurements of the procedure were measured on the LPC55S69 running at 150 MHz. To simplify the connection to the Verifier, a serial connection was established using the on-chip UART configured to a baud rate of 806,400. The LPC55S69 hash engine was used to compute the necessary authentication using SHA-256 for hashing and the HMAC. The procedure was tested on different memory sizes, increasing in steps of 1 KB. The memory offloaded was the 240 KB of non-secure RAM associated with the IoT application.

The time used for the offloading procedure is proportional to the offloaded memory size, as shown in Figure 15. Furthermore, the time used for memory authentication scales the memory size, but it is negligible compared to the time required by data transmission. The duration also scales w.r.t. size but is slightly noisy and requires at least 0.23 ms, as shown in Figure 16. In Figure 16, the similarity to a step function is due to the processor’s internal hash engine. The maximum input we give to the hash-engine is 32 KB, this means that memory chunks larger than 32 KB requires multiple rounds of the hash-engine to be processed. This results in the engine handling inputs of 0–32 KB, 32–64 KB, etc., in approximately the same time frame.

This offloading RA approach can be combined with current methods of static attestation through program checksum comparison. In that case, the code memory (and boot ROM) can be attested through hash comparisons, while offloading the dynamic memory: the RAM, any data region of the flash/EEPROM and internal/external peripheral registers. When combining these methods on the aforementioned IoT sensor application, offloading the RAM, external peripheral registers, and the I2C peripheral, while only transmitting the hash of the flash and boot ROM, the entire process takes 3.94 s using the previously specified hardware and communication setup. The experimental results are overall comparable with other RA schemes [53] and confirm the feasibility of ERAMO.

### 10.1. Energy Consumption

In this section, we provide the energy consumption measurements for the offloading procedure. Voltage is measured over the shunt resistor (resistance of shunt resistor R = 2.43 ohm) on the evaluation board (during normal operations and during offloading). This is a method suggested by the LPC55S69 evaluation board user manual [57]. Having the voltage and resistance allows us to calculate the power: $P=\frac{V^{2}}{R}$. Comparing the power during offloading to the power during normal operations allows us to calculate the excess power usage caused by running ERAMO. The duration of the offloading procedure is also measured. Having both the power usage and the duration allows us to calculate the energy usage: E=P·t. Calculating the energy usage with the excess power allows us to calculate the excess energy usage caused by running ERAMO. The results are summarized in Table 2. The power consumption of the offloading is 208 mW and takes 23.3 s at a baud rate of 460,800. Given that the normal power consumption is 190 mW, the excess power will be 18 mW, resulting in an excess energy consumption of 419 mJ (116 μWh). This does however vary with baud rate as the process will be longer and thus more energy consuming at lower baud rates, an excess 16 mW was measured at 115,200 with the procedure taking 89 s, resulting in an excess 1424 mJ, more than three times as much the procedure at a baud rate of 460,800.

Note that the power measured is the power used by the entire evaluation kit (MCU, DC-DC converter, IO etc.) and would be lower if a simpler device was produced including only the necessary hardware. However, hardware used for wireless communication would in turn increase the energy consumption.

### 10.2. Verification Process: Fog Node’S Perspective

During the offloading procedure, the fog node, acting as the Verifier, performs the verification. Listing 1 shows a simplified example for the verification where for each memory region processed, its hash is computed and compared to its previous hash, and the information regarding the offloading is written to the terminal.

**Listing 1.** The Verifier receiving and attesting the offloaded memory.

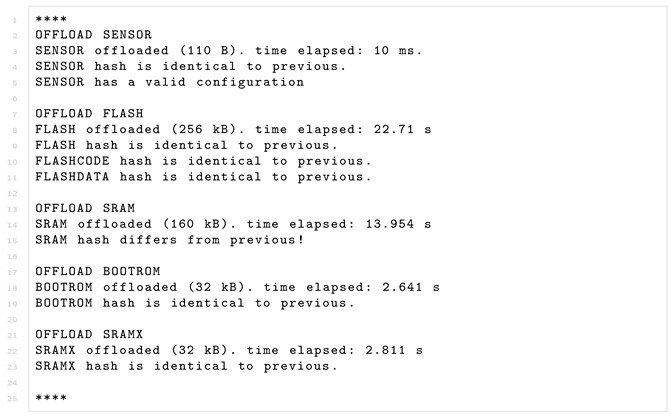



However, attestation of a sensor peripheral necessitates a more complicated verification than a simple hash comparison. For example, we tried to verify the integrity of the peripheral temperature sensor while also offloading the sensor’s registers. Since the microcontroller does not have direct access to these registers, it starts the procedure by initiating an I2C burst read starting from the first mapped register, 0x88 the *calibration* register, to the last, 0xF5 the *configuration* register, following the previously-described sensor memory map (depicted in Figure 12). As a result, 110 byte-sized registers are offloaded. Once the MCU has read the data using an I2C burst read, it is then sent byte-by-byte to the fog node. To verify the state of the temperature sensor, we construct a white-list of allowed data values per register. This is a relatively simple way for performing verification, but it serves as a proof-of-concept of verification without the use of hashes. The white-list is formatted to have the address followed by the allowed values as shown in Listing 2.

**Listing 2.** Register white-list for digital temperature sensor.

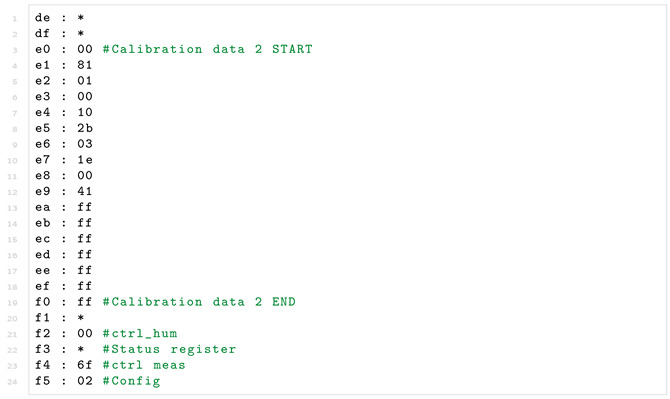



The verification program goes byte-by-byte through the file comparing each value of the sensor to the white-listed value. If a value at a specific address is not on the white-list, the program will flag the sensor as having an invalid configuration. Such a simple approach could be also employed to check certain memory addresses of the RAM against a white-list. For instance, to check if small memory areas, such as the FreeRTOS task definitions and boundaries remain static and whether predictable regions, such as the queue-structure contain values according to those expected of the device.

## 11. Eramo Limitations

The current design of the ERAMO protocol poses some limitations in dealing with various types of attacks. In particular, the challenge-response design of the protocol increases the risk of Denial of Service (DoS) attacks. For instance, the attacker may impersonate the Verifier and send frequent attestation requests to the Prover to suspend the regular operation of the Prover. To reduce the DoS risk, the design can be extended by replacing the challenge-response protocol with a non-interactive approach (such as [26,58]), in which the device self-initiates attestation through a secure algorithm running on a secure memory (e.g., TrustZone).

Additionally, ERAMO approach is susceptible to mobile adversaries, Specifically, the malware which may delete itself or relocate to another device to avoid detection is referred to as *transient malware*. The malware that moves to a different memory region on the same device to avoid detection is called *migratory malware*. To secure the protocol against transient and migratory malware, memory locking (described in Section 12) can be used to accurately reflect the state of the memory region at a given point in time.

## 12. Discussion

ERAMO approach opens the possibilities of various RA schemes, such as allowing ranges of values, complex combinations of settings or even using machine learning to determine the validity of the dynamic memory. Due to the increased computational power of the Verifier, memory forensics tools [37] may be used on the memory dump allowing the Verifier to distinguish a legal state of memory from exploited memory.

This paper presents comprehensive implementation details of the ERAMO protocol on a TrustZone-capable LPC55S69 running an IoT sensor application on FreeRTOS. However, at a large extent, this implementation is compatible with other IoT platform configurations. For instance, the proposed protocol can be migrated to a Raspberry pi 3 or 4 Model B that supports OP-TEE [59] as an operating system in the TrustZone environment and Linux operating system running on the “normal world”. To the best of our knowledge, we are not aware of an integration of TrustZone with popular IoT mobile operating systems such as Contiki, TinyOS, RIOT. However, if there exist such an implementation, our proposed technical solution should be compatible with a large-scale linux-based mobile operating system running on the “normal world”.

To mitigate the downtime caused by attestation, ERAMO can utilize TrustZone’s seperation between the secure and non-secure domain and allow the device and its OS to resume its normal operation during attestation. This however introduces the possibility of mobile adversaries, which must be countered. While ERAMO aims to improve mobile adversary detection by attesting many memory regions, still during the attestation, a mobile adversary may evade detection by relocating itself in different memory blocks within one memory region. If the trusted component on the Prover deploys a *memory locking* technique, it would be possible to guarantee the result’s integrity while allowing the execution of regular operations to run simultaneously with the attestation. Memory locking is already used in Linux to lock memory pages in RAM. Recently, it has been proposed for achieving temporal consistency in embedded systems [60]. The different methods of memory-locking and their consequences regarding temporal consistency and vulnerabilities are listed below:No-Lock: A naive solution to this issue is to let the attestation procedure run concurrently with the devices normal operation without locking. This does not ensure any temporal consistency, leaving the system vulnerable to mobile adversaries.All-Lock: All-Lock is the contrary to the No-Lock and avoids the mobile malware. It instead keeps the system temporally consistent throughout the entire process. This method is equivalent to having an uninterruptible attestation procedure.Decreasing Lock (Dec-Lock): Like All-Lock, this method detects mobile adversary, while having the benefit of the memory being gradually unlocked.Increasing Lock (Inc-Lock): Inc-Lock does provide temporal consistency at the time which the attestation. This will be enough to detect migratory malware, however it does not protect against mobile malware. Malware not located in the first region to be attested could delete itself before the memory is locked and could thereby avoid detection.Copy Lock (Cpy-Lock): Cpy-Lock protects against mobile malware but is only viable if the time required to copy is less than time required for the attestation.Extended locking: Extended variants All-Lock-Ext and Inc-Lock-Ext can be used instead of their normal counterparts if consistency is needed until the memory has been attested by the verifier. This is done by locking the memory until the attestation is finished, which is useful to prevent TOCTOU vulnerabilities.

To this end, the memory-locking technique can be a promising technique in ensuring the temporal consistency of the offloaded memory. Through memory locking, it is possible to lock a memory block, preventing it from being changed until it is again unlocked. This can be used to lock the memory before being attested and gradually unlock it as soon as it is offloaded or attested, following the Dec-Lock approach described above. Consequently, any malware or effects caused by malware and memory exploits will be locked in memory until it is offloaded, causing it to be detected by the Verifier. Furthermore, TrustZone will ensure that any malware present on the non-trusted component cannot interfere with the trusted offloading procedure, ensuring that the offloading procedure can run securely simultaneous with Prover’s regular operations. As an example, using Dec-Lock, the memory areas can be offloaded and unlocked in the following order:1.The memory area of the main stack is offloaded and unlocked.2.The memory area of the main heap (before the allocated task) is offloaded and unlocked.3.The memory area of the *Idle task* and software timers are offloaded and unlocked.4.The memory area of the scheduler is offloaded and unlocked.5.Start allowing interrupts from the application.6.The memory area of the queue for the transmissions task is unlocked (e.g., UART Transmit Queue).7.The memory area of the communication transmission task is offloaded and unlocked (e.g., UART Transmit Task).8.The communications task is resumed.9.The memory area of the temperature sensor task is offloaded and unlocked.10.The temperature sensor task is resumed.

Following this approach, the critical tasks are unlocked early than the less important tasks before moving on to the flash memory and other static areas. In this way, the device can quickly resume its most critical functions, while the rest of the memory is analyzed. This may especially minimise downtime when the device has a lot more memory available than what is being used by the application.

## 13. Conclusions and Future Works

This paper presented ERAMO, a novel RA protocol that relies on a memory offloading approach to verify the Prover’s integrity. ERAMO allows the verification of more dynamic memory areas (such as the internal and external peripherals) which are not covered by existing RA schemes. In addition, we provide all the experimental details for memory analysis in a multi-service device, showing that it is possible to predict their memory allocation. We implemented the hardware proof-of-concept using an ARM Cortex-M33-based microcontroller with ARM TrustZone support. The performance analysis regarding essential metrics such as transmission time, data authentication time, and energy efficiency clearly shows its effectiveness.

As future work, we plan to perform a complete implementation and evaluation of the protocol including the fog node’s perspective. In particular, this enhancement will include the implementation of embedded application memory analysis performed by the fog node and more extensive evaluation regarding the overall attestation time. We will also extend the multi-service nature of the proposed protocol into an approach that attests large number of interacting services in IoT swarms. We plan to implement the proposed memory locking scheme and evaluate it w.r.t. its effects on the efficiency of the attestation procedure. Furthermore, we will explore different approaches to optimize cryptographic details of the proposed attestation protocol. Finally, future work includes also implementing the Verifier side, including forensic tools on the RAM and peripherals.

## Figures and Tables

**Figure 1 sensors-22-04340-f001:**
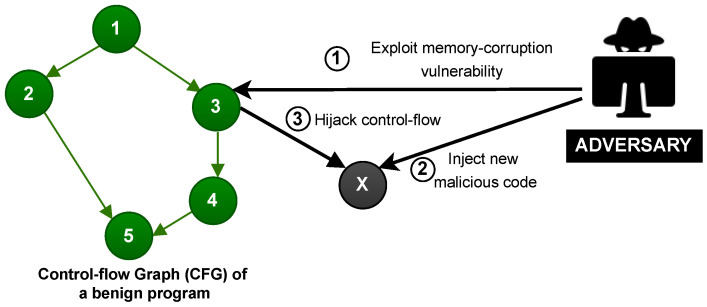
Code-injection attack. The Control Flow Graph (CFG) represents the legitimate execution flows of a benign software, where each graph node (Nodes 1–5) denotes a software instruction. A code-injection adversary performs the actions 1–3.

**Figure 2 sensors-22-04340-f002:**
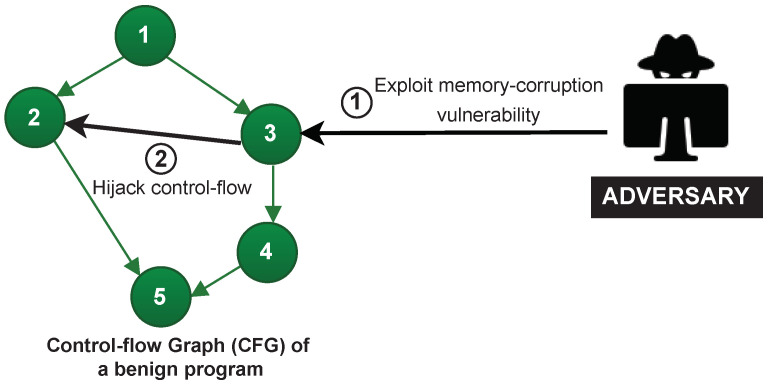
Code-reuse attack. The Control Flow Graph (CFG) represents the legitimate execution flows of a benign software, where each graph node (Nodes 1–5) denotes a software instruction. A code-reuse adversary performs the actions 1–2.

**Figure 3 sensors-22-04340-f003:**
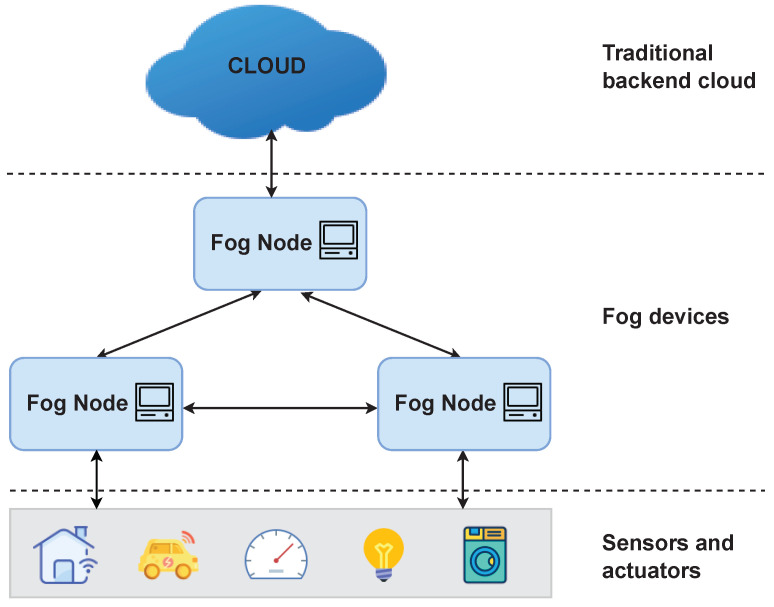
Fog Computing paradigm.

**Figure 4 sensors-22-04340-f004:**
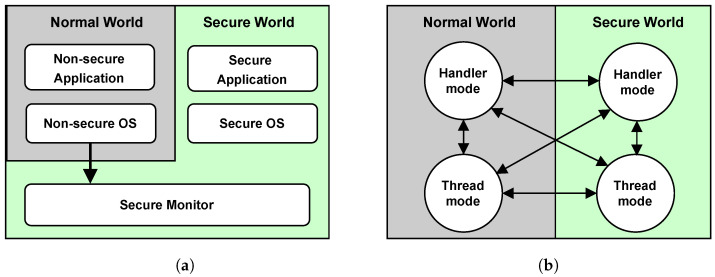
Differences between TrustZone on Cortex-A and Cortex-M. (**a**) TrustZone Cortex-A. (**b**) TrustZone Cortex-M.

**Figure 5 sensors-22-04340-f005:**
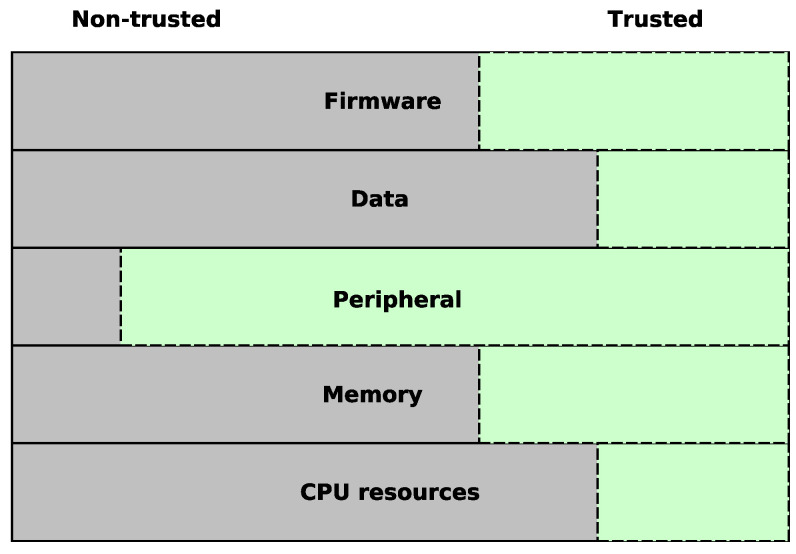
TrustZone view of a system address space and resources.

**Figure 6 sensors-22-04340-f006:**
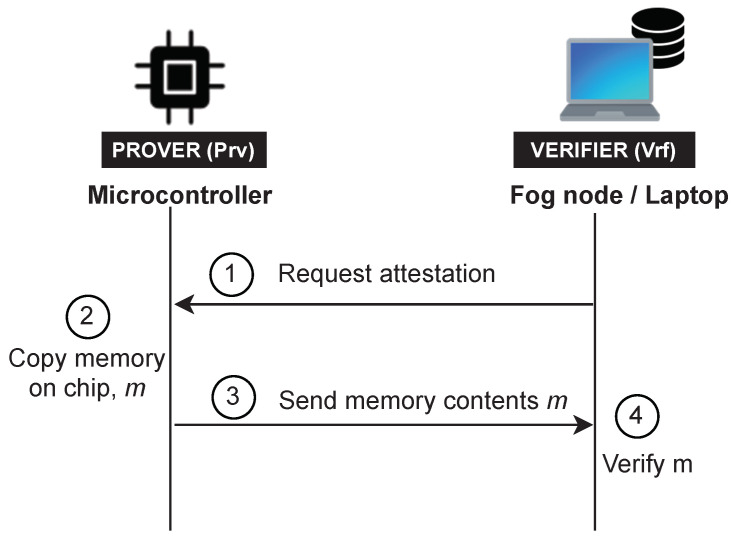
System model of the memory offloading protocol.

**Figure 7 sensors-22-04340-f007:**
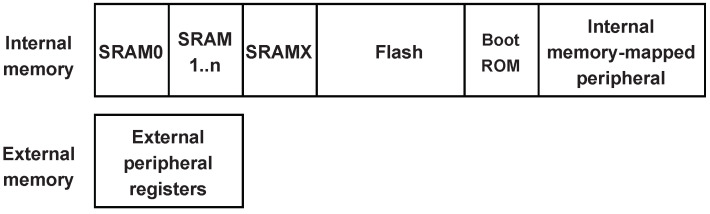
Memory on chip of a Prover device.

**Figure 8 sensors-22-04340-f008:**
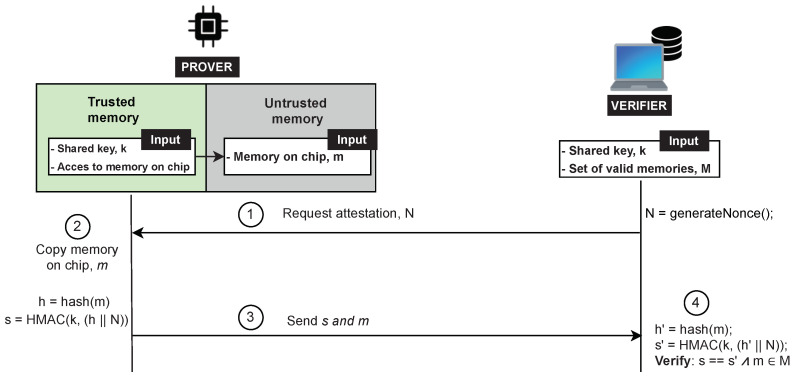
ERAMO protocol.

**Figure 9 sensors-22-04340-f009:**
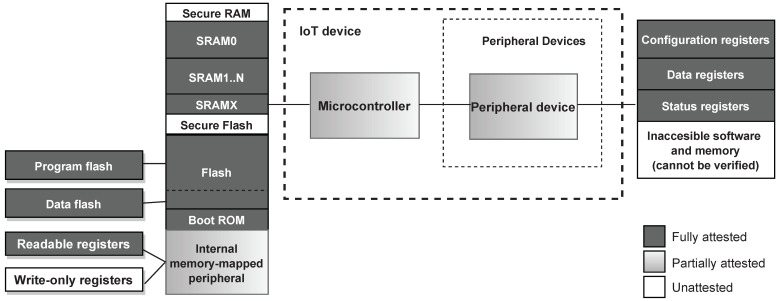
Memory regions attested by the ERAMO protocol.

**Figure 10 sensors-22-04340-f010:**
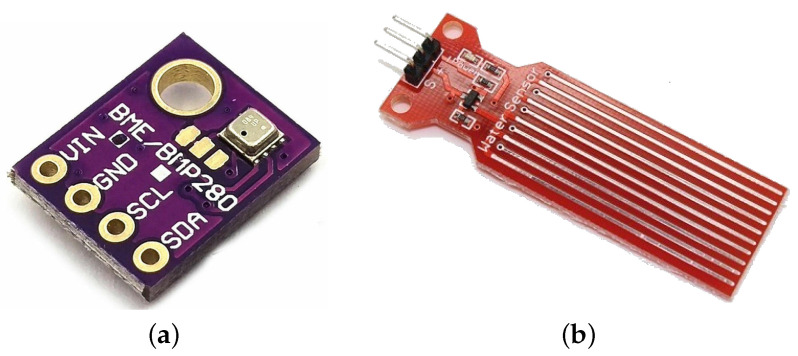
The two sensors used for the sensor node application. (**a**) Digital temperature sensor. (**b**) Analog water level sensor.

**Figure 11 sensors-22-04340-f011:**
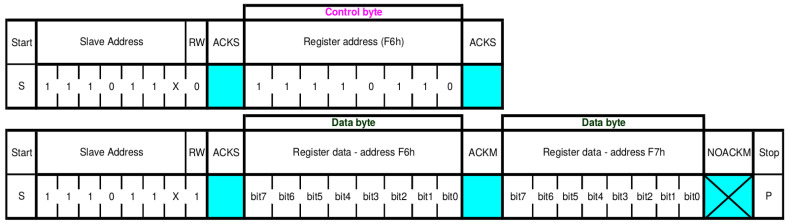
Sensor I2C burst read of register 0xF6 and 0xF7 [54].

**Figure 12 sensors-22-04340-f012:**
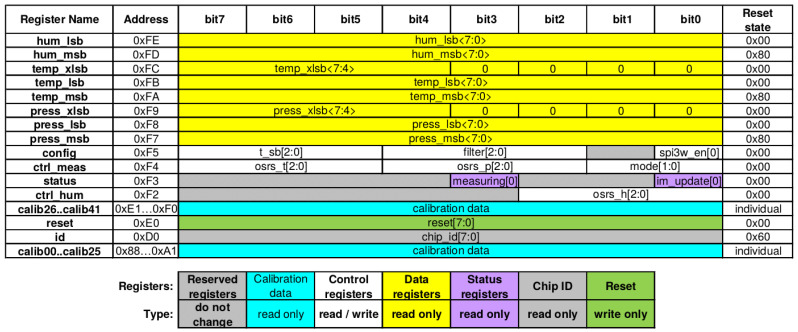
Temperature sensor BME280 memory map [54].

**Figure 13 sensors-22-04340-f013:**
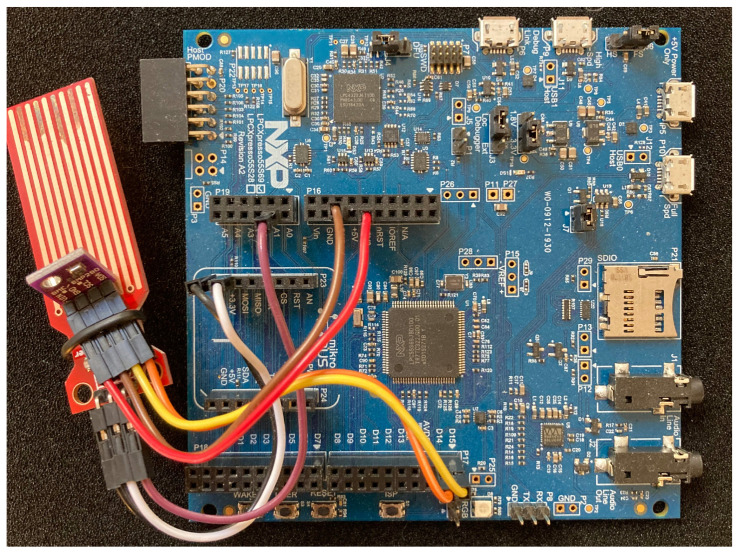
IoT device setup consisting of the TrustZone-enabled LPC55S69 evaluation board and the two connected sensors.

**Figure 14 sensors-22-04340-f014:**
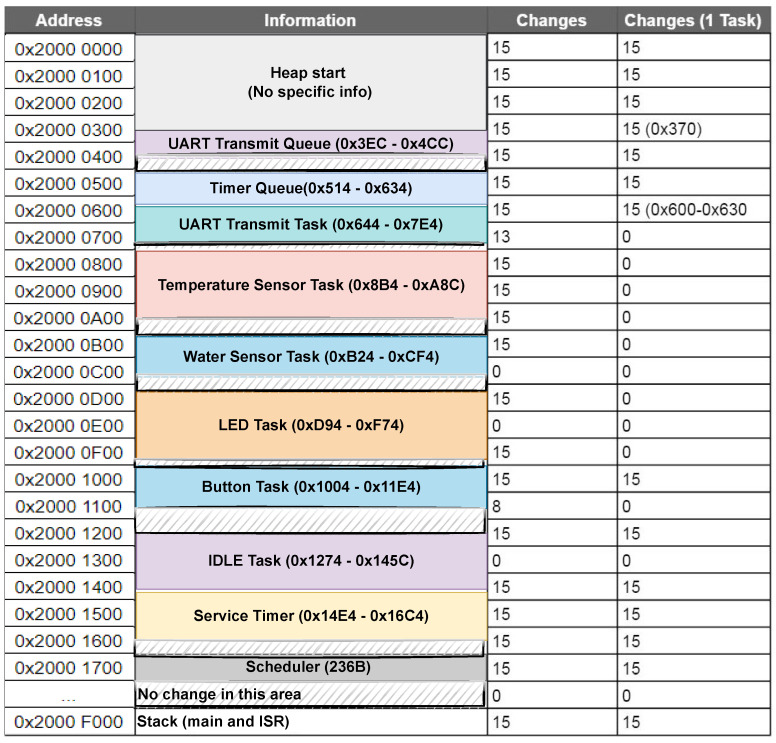
SRAM0 memory map with amount of changes between 16 memory offloads.

**Figure 15 sensors-22-04340-f015:**
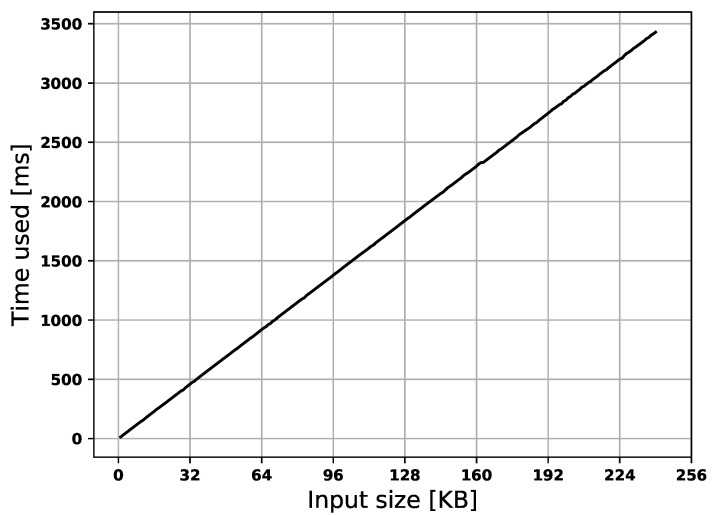
Time used to transmit memory.

**Figure 16 sensors-22-04340-f016:**
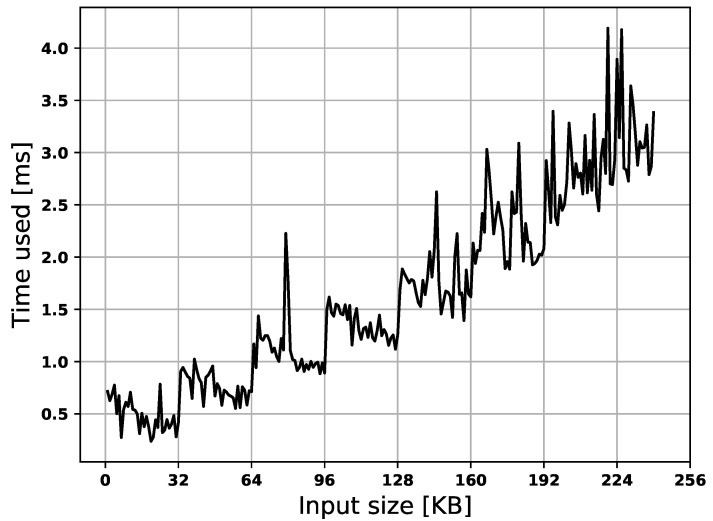
Time used to authenticate memory.

**Table 1 sensors-22-04340-t001:** Related work summary.

Scheme	Static Memory	RAM	Peripheral	Verification	Type	Attestation
SWATT [13], Pioneer [14]	●	○	○	Program checksum	One-to-one	On-demand
SMART [19], TrustLite [20], TyTan [21]	●	○	○	Program checksum	One-to-one	On-demand
C-FLAT [5], LO-FAT [6]	○	◐	○	Control flow integrity (CFI)	One-to-one	On-demand
ATRIUM [7], LiteHAX [28]	●	◐	○	Program checksum & CFI	One-to-one	On-demand
SMARM [9]	●	○	○	Program checksum & Shuffled Measurements	One-to-one	On-demand
ERASMUS [10]	●	○	○	Program checksum	One-to-one	Self-initiated
SEDA [22], SANA [23], SARA [33]	●	○	○	Program checksum	One-to-many	On-demand
DIAT [30]	●	◐	○	Program checksum & CFI	Many-to-many	On-demand
RADIS [31], ARCADIS [34]	○	◐	○	Program checksum & CFI	One-to-many	On-demand
CloneCloud [35]	○	●	○	○	One-to-one	○
**ERAMO**	●	●	◐	**Memory offloading**	**One-to-one**	**On-demand**

**Table 2 sensors-22-04340-t002:** Energy usage in ERAMO for different communication rates.

	Communication Rate [bytes/second]	Voltage over Shunt Resistor [mV]	Power [mW]	Excess Power [mW]	Duration of ERAMO Offloading [s]	Excess Energy [mJ]
Normal operation (no offloading)	NAN	679	190	NAN	NAN	NAN
Offloading using ERAMO	460,800	711	208	18	23.3	419
Offloading using ERAMO	115,200	708	206	16	89.0	1424

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
