# Peer review of "Memory Offloading for Remote Attestation of Multi-Service IoT Devicesâ€"

_sensors, 2022, doi:10.3390/s22124340_

Round 1
Reviewer 1 Report
The article addresses issues of IoT security through investigation of remote attestation mechanisms. Remote attestation is regarded as a prospective mechanism for organizing defense of IoT multi-service devices and detecting some specific malware, especially located in program memory.
The analysis of related work is well written and covers quite significant references both well known papers and fresh ones.
A key element of the work is the use of fog computing. As it is stated with the help of fog computing the Verifier is implemented in a form of a fog node. However, the paper does not clearly enough show that specificity of this technology, which is directly used within the framework of the proposed ERAMO protocol. In addition, the use of fog computing is not sufficiently represented in the article.
Section 2 gives a generalized understanding of the target type of attack. Section 6.1 gives 4 adversary characteristics (adversary model). Several particular ways of attacks are also mentioned there, such as Denial of Service and Time-Of-Check Time-Of-Use. However, the work lacks a full-fledged attack model, which would reveal the main characteristics of relevant attacks, such as goals, steps, restrictions, resources, etc.
The Evaluation section lacks an explanation of the reason for the occurrence of peaks and drops in memory authentication time, depending on the size of the input (Figure 13). Indeed, this time is negligible compared to the memory transmit time, but nevertheless I would like to understand why it differs so much from the linear one. If possible, I would also like to see more detailed data (diagrams or tables) in section 9.1 for assessing energy consumption. The Evaluation section lacks any evaluation of the proposed protocol for its susceptibility to attacks.
The Discussion section could be expanded. I would like the authors to explain if their proposed technical solutions are portable to other mobile operating systems of IoT devices.
Author Response
1.1 Reviewer#1, Concern #1: A key element of the work is the use of fog computing. As it is stated with the help of fog computing the Verifier is implemented in a form of a fog node. However, the paper does not clearly enough show that specificity of this technology, which is directly used within the framework of the proposed ERAMO protocol. In addition, the use of fog computing is not sufficiently represented in the article.
Author response: We thank the reviewer for pointing out the concern.
Author action: We have included a discussion of the Fog computing paradigm in Section 4.2. Additionally, we also expanded the system model in Section 5 to clarify the usage of fog computing in the proposed ERAMO protocol.
1.2 Reviewer#1, Concern #2: Section 2 gives a generalized understanding of the target type of attack. Section 6.1 gives 4 adversary characteristics (adversary model). Several particular ways of attacks are also mentioned there, such as Denial of Service and Time-Of-Check Time-Of-Use. However, the work lacks a full-fledged attack model, which would reveal the main characteristics of relevant attacks, such as goals, steps, restrictions, resources, etc.
Author response: We thank the reviewer for the comment. We do agree that the attack model needs further clarification.
Author action: We have rearranged Section 6 for better readability. In particular, we have expanded “Attack capabilities and limitations” in Section 6.2 and we clarified adversarial capabilities. In addition, we have added Section 4.1 to clarify the actions and goals of runtime attacks. To this end, we have added Figure 1 and Figure 2.
1.3 Reviewer#1, Concern #3: The Evaluation section lacks an explanation of the reason for the occurrence of peaks and drops in memory authentication time, depending on the size of the input (Figure 13). Indeed, this time is negligible compared to the memory transmit time, but nevertheless I would like to understand why it differs so much from the linear one.
Author response: The occurrence of peaks and drops is due to the processor's internal hash engine. The maximum input we give to the hash-engine is 32KB, this means that memory chunks larger than 32KB require multiple rounds of the hash-engine to be processed. This results in the engine handling inputs of 0-32KB, 32-64KB etc. in approximately the same time frame.
Author action: We have included the clarification in Section 10.
1.4 Reviewer#1, Concern #4: If possible, I would also like to see more detailed data (diagrams or tables) in section 9.1 for assessing energy consumption.
Author response: We thank the reviewer for the comment.
Author action: We revised Section 10.1 and included a detailed energy consumption of ERAMO w.r.t different communication rates. In particular, we added Table 2 to summarize the energy usage in the proposed protocol.
1.5 Reviewer#1, Concern #5: The Evaluation section lacks any evaluation of the proposed protocol for its susceptibility to attacks.
Author response: We thank the reviewer for the comment.
Author action: We have added Section 11 with limitations of ERAMO regarding susceptibility to attacks, such as DoS and mobile adversary, presented in the Threat model (Section 6).
1.6 Reviewer#1, Concern #6: The Discussion section could be expanded. I would like the authors to explain if their proposed technical solutions are portable to other mobile operating systems of IoT devices.
Author response: We thank the reviewer for this insightful comment.
Author action: The proposed technical solution is interoperable with other IoT platforms and we have clarified the compatibility of ERAMO in the Discussion (Section 12).
Reviewer 2 Report
The paper presented in this journal has tackled a very interesting topic, the issue of remote attestation is a quite novel topic and the proposal made by the authors to make use of other nodes to perform this action is also very promising.
Nevertheless, after comparing both papers (conference and this one), I think that a more extensive evaluation should be performed. A comparison with other alternatives will provide these means so as to see the efficiency of this proposal.
Additionally, the authors include the use of fog nodes to perform the verification. So, why not to provide an architecture to identify different nodes also in different planes (edge/fog/cloud). Moreover, why not include the advantage of fog computing in this architecture?
in my opinion, a more extensive evaluation will also be fundamental to extend the research of this proposal. A more fair comparison where the whole process performance is assessed should be presented, not only analyzing the time to transmit memory or to authenticate it. I assume the this RA should outperform other works in the literature, but there is no section probing it.
Author Response
2.1 Reviewer#2, Concern #1: The paper presented in this journal has tackled a very interesting topic, the issue of remote attestation is a quite novel topic and the proposal made by the authors to make use of other nodes to perform this action is also very promising.
Author response: We thank the reviewer for the positive comments on our manuscript and appreciation of our work.
2.2 Reviewer#2, Concern #2: Nevertheless, after comparing both papers (conference and this one), I think that a more extensive evaluation should be performed. A comparison with other alternatives will provide these means so as to see the efficiency of this proposal.
Author response: We thank the reviewer for the comment. Indeed, we agree that a comparison w.r.t. the state-of-the-art solutions will be beneficial to understand the security and efficacy of ERAMO protocol. In terms of security, ERAMO attests dynamic memory regions and peripherals of IoT devices. Such attacks remain undetected by existing RA schemes. In terms of performance, it is challenging to perform an empiric performance comparison between different attestation schemes. In particular, various changes of an environmental setup such as customized hardware proof-of-concept implementations, different configurations for the simulations, and non-available source code do not allow the reproduction of the same results.
Author action: Unfortunately, we could not compare the performance of ERAMO w.r.t. the state-of-the-art RA protocols. However, to improve the overall evaluation of the proposed protocol we incorporated more details in the energy consumption (Section 10.1) and added the preliminary evaluation of the verification process (Section 10.2).
2.3 Reviewer#2, Concern #3: Additionally, the authors include the use of fog nodes to perform the verification. So, why not to provide an architecture to identify different nodes also in different planes (edge/fog/cloud). Moreover, why not include the advantage of fog computing in this architecture?
Author response: We thank the reviewer for pointing out the concern. We agree with the comment. For simplicity, we consider a system with only one layer with fog devices, assuming that the Verifier is a fog node with powerful resources and computational capabilities.
Author action: We expanded the system model in Section 5 to clarify the usage of fog computing in the proposed ERAMO protocol. We have also provided a discussion of the Fog computing in Section 4.2 and included Figure 1 for a better overview of the fog computing paradigm.
2.4 Reviewer#2, Concern #4: In my opinion, a more extensive evaluation will also be fundamental to extend the research of this proposal. A more fair comparison where the whole process performance is assessed should be presented, not only analyzing the time to transmit memory or to authenticate it. I assume the this RA should outperform other works in the literature, but there is no section probing it.
Author response: We thank the reviewer for the comment. The main objective of this work is to investigate the memory offloading approach from the IoT device perspective. For this reason, we evaluated only the transmission and authentication that are both performed by the IoT device. A complete performance evaluation would first require the implementation of embedded
application memory analysis by the fog node, and then a more extensive evaluation regarding the attestation time. However, since our study already explores numerous fundamental and preliminary issues about the memory offloading for remote attestation, we will have to investigate such more advanced aspects in our future research. Moreover, in the future, we plan
to implement the memory locking mechanism and integrate with the current TrustZone implementation, allowing the application to run while its being attested, thus, making the time used on attestation less relevant.
Author action: We have introduced a new Section 10.2 with preliminary evaluation of the verification process from the fog node’s perspective, including two approaches: hash comparison (Listing 1) and white-list comparison (Listing 2). The implementation of embedded application memory analysis along with complete performance evaluation have been added as future works in Section 13.
Round 2
Reviewer 1 Report
The specificity of the proposed solution has been added and clarified in context of ERAMO protocol.
Sections 4.1 and 6.2 have been significantly expanded by adding extra clarification on relevant attacks. However it still lacks some more systematic and complete view on all possible relevant attacks here, not only two mentioned types, code-injection attack and code-reuse attack.
An explanation of peaks and drops in memory authentication time has been given. Evaluation of energy consumption has been presented, concrete numerical data given in a table.
In discussion possibility to migrate the solution to some other operating systems (mainly Raspberry pi 3 or 4 Model B) have been disclosed, but it'd be better to state what and in what way should be adopted to facilitate such porting to such RPi 3/4B.
Author Response
Reviewer#1, Concern #1: Sections 4.1 and 6.2 have been significantly expanded by adding extra clarification on relevant attacks. However it still lacks some more systematic and complete view on all possible relevant attacks here, not only two mentioned types, code-injection attack and code-reuse attack.
Author response: We thank the reviewer for pointing out the concern. A systematic and complete view on all possible relevant attacks would surely be valuable, but we think it would require / end up in another paper. This is indeed a great idea for a survey paper! For this paper, we have decided to stick to the assumptions that are well known and used in RA literature, also to have a common ground to compare our proposal to the several protocols in literature.
Author action: We have expanded Section 6.2 to clarify that, in this paper, we consider the most
relevant attacks as per RA literature. Thus, a systematic and complete view on all possible
relevant attacks to RA protocols is out of the scope of this paper.
1.2 Reviewer#1, Concern #2: In discussion possibility to migrate the solution to some other operating systems (mainly Raspberry pi 3 or 4 Model B) have been disclosed, but it'd be better to state what and in what way should be adopted to facilitate such porting to such RPi 3/4B.
Author response: We thank the reviewer for the suggestion on how to improve the paper.
Author action: We have extended Section 12 to give some additional technical details how our
technical solution can be migrated to other IoT platforms.
Reviewer 2 Report
Although the authors of the paper did not fully address the comments made in the previous review. I think that the paper has been notably evolved and it now meets the quality requirements of this journal.
I see no inconvenience in accepting this paper.
Author Response
Reviewer#2, Comment #1: Although the authors of the paper did not fully address the comments made in the previous review. I think that the paper has been notably evolved and it now meets the quality requirements of this journal. I see no inconvenience in accepting this paper.
Author response: We thank the reviewer for the positive feedback.